# Psychosocial Distress and the Quality of Life of Cancer Patients in Rural Hospitals in Limpopo Province: A Qualitative Study

**DOI:** 10.3390/curroncol32010043

**Published:** 2025-01-16

**Authors:** Dorah Ursula Ramathuba, Neo Jacqueline Ramutumbu

**Affiliations:** Department of Advanced Nursing Science, University of Venda, P/Bag X5050, Thohoyandou 0950, South Africa; nramutumbu@gmail.com

**Keywords:** cancer, cancer patients, biopsychosocial symptoms, psychosocial distress, uncertainty, quality of life

## Abstract

Background: The diagnosis and treatment of cancer are associated with substantial physical, psychological, and social morbidity for most patients. Distress can be seen as an unpleasant experience of an emotional, psychological, social, or spiritual nature that interferes with the ability to cope with cancer treatment. Purpose: The aim was to understand patients’ experiences of distress in their context and to analyze and interpret the findings. Method: An explorative, descriptive qualitative study was conducted among cancer patients receiving treatment and care at rural hospitals in Limpopo. A face-to-face individual interview was conducted to determine the participants’ cancer-related experiences and quality of life. Thematic analysis was conducted following Tesch’s method, and the themes developed were subjected to a triangulation process to ensure the validity and rigor of the findings. Findings: The participants revealed experiences of symptomatic distress resulting in biopsychosocial distress such as pain, fatigue, emotional distress related to prognosis and uncertainty about the future, psychosocial distress related to a lack or absence of support, financial instability, and poor self-esteem. Conclusions: Cancer patients face many challenges during their treatment journey. Participants were drained by anxiety and uncertainty of the cancer trajectory and required psychosocial support. The oncology team must provide supportive preventive measures for side effects management and culture-sensitive psychotherapy at an early stage to improve their quality of life.

## 1. Introduction

Cancer is common worldwide, and the cancer burden is disproportionately concentrated in low- and middle-income countries (LMICs) and is among the prime causes of mortality and morbidity, especially where oncology services are absent or inaccessible due to poor or a lack of specialized oncology services. According to the World Health Organization (WHO) International Agency for Research Centre’s Report 2020, the global cancer burden was estimated to have risen to 19.3 million new cases and 10 million deaths [1].

The World Health Organization quality of life assessment (WHOQOL) tool measures the physical function, mental status, and ability to engage in normative social interactions of patients experiencing chronic and debilitating diseases [2]. Oncological professionals have attempted to close gaps in the psychosocial care of cancer patients and have identified the use of biopsychosocial assessment tools in improving the coordination of integrated treatment efforts between medical and mental healthcare providers [3]. However, these tools cannot identify significant leverage points in treatment planning for biological, social, and psychological health determinants because the underlying individual, administrative, and structural factors need to be understood in cancer care, especially in the rural areas of Limpopo province. 

South Africa (SA) has poor health outcomes in both rural and urban areas; however, rural communities experience significant barriers to accessing healthcare, including financial barriers, inadequate transport, and distance to the nearest facility and the package of services available at different levels of care, of which oncology services are absent at the PHC and district levels. The availability of specialist services at regional hospitals may vary, and patients from rural hospitals must be referred to tertiary services, usually even further removed geographically [4]. Therefore, patients experience language barriers and culture shock and cannot adjust to an unfamiliar setting when referred to tertiary services. The biopsychosocial model is an approach to understanding mental and physical health through a multi-systems lens of biology, psychology, and social environment. It emphasizes the unified and interactive roles of biological, psychological, and social factors in understanding how these systems overlap and interact to impact everyone’s well-being and risk for illness, and understanding these systems can lead to more effective treatment. Distress in cancer patients is expected, given the health and social changes a cancer diagnosis brings. It can be seen as an unpleasant experience of an emotional, psychological, social, or spiritual nature that interferes with the ability to cope with cancer treatment. Wang and Feng [5] indicate that as a chronic disease, the diagnosis and treatments of cancer cause patients to be vulnerable to changes in their self-concept and perceptions.

Biopsychosocial factors associated with quality of life represent a complex set of variables that impact an individual’s physical, emotional, social, and psychological well-being. Physical symptoms cause physical distress as the patients are unable to carry out everyday activities like bathing, eating, and other chores due to side effects from cancer treatment. Emotional distress results from the emotional pain, uncertainty, and fear of death as rural communities associate cancer with death. Psychosocial distress results from poor social support, a lack of social networks, and a lack of healthcare providers’ support, which has a negative impact on their quality of life. Cultural taboos and myths surrounding cancer can further aggravate the distress in patients as cancer is regarded as fatal, and the associated stigma surrounding cancer treatment can result in a patient’s withdrawal, isolation, and lack of social support networks since communities associate it with HIV. Savas and Ozem [6] indicate that cancer causes adverse effects on daily activities and social communication and can lead to isolation, vulnerability, withdrawal, or the harmful transmission of emotions.

Ebob-Anya and Bassah [7] suggest that to ensure a good quality of life, cancer patients must receive multidimensional care that encompasses physical, social, psychological, and spiritual domains.

Biopsychosocial models are appropriate as they intertwine the biomedical and psychosocial impacts of cancer but may not be appropriately understood if only assessed quantitatively. Most of the literature and studies have assessed the quality of life of cancer patients quantitatively, and little has been conducted to explore how they perceive, value, and experience the quality of their lives to fill the gap in a rural setting. Therefore, this study aims to explore the psychosocial problems cancer patients experience in-depth and their reflections during the treatment process of the cancer disease.

### Setting

South Africa has a two-tiered and highly unequal healthcare system. The public sector is state-funded and caters to the majority (71%) of the population while 27% have medical insurance and use the private sector [8]. The public sector has three levels of care starting at local municipalities with primary healthcare facilities with not enough professional registered nurses; the district hospitals that are also ill-equipped and have general practitioners with no specialist care; and the provincial hospitals where there is a lack of specialists and advanced technology. The funding allocated to the public health sector is further distributed to eight provinces, with Limpopo being one where the population demands exceed the care. Cancer patients from Limpopo’s rural districts must travel to provincial hospitals for further diagnosis and treatment and to academic hospitals outside the province. Most patients delay access to medical care; sometimes, care is interrupted due to the absence or commitments of medical oncologists and radiation oncologists. In the province, there is only one oncologist who consults all referrals. Most rural communities live below the poverty line and rely on government grants (old-age pension fund); the unemployment rate is at 33.5% [8]. Therefore, the impact of a cancer diagnosis has a severe impact on the quality of their life as patients have to travel for treatment, return to primary health settings in districts where there is minimal care and support due to the lack of specialist care, social support, and health providers care resulting in distress.

## 2. Methodology

A qualitative explorative, descriptive, and phenomenological study was conducted among cancer patients receiving treatment and care at rural hospitals in Limpopo using semi-structured, audio-recorded interviews between March and May 2019. A phenomenological inquiry approach was appropriate to this study to explore and characterize participant patients’ lived experiences and shared meaning. The objective is to explore and describe how cancer treatment affects their physical, social, and psychological well-being. The rationale for selecting this type of research design is based on the interest in collecting the subjective and personal views of cancer patients in a natural clinical setting. The aim is to obtain a straight description of the patients from rural districts with no oncology centers and limited high-technology resources to understand experiences to carry out an analysis and interpretation of the findings. 

### 2.1. Data Collection

Data collection methods are techniques and procedures used for gathering information for research purposes [9]. The population was cancer patients receiving cancer treatment, and a purposive sampling method was used to sample patients who were receiving cancer treatment for one year or more. All cancer patients, irrespective of gender, type of cancer, and treatment, were included in the study; those who were still in the process of finalizing the diagnosis and at stage 1 of cancer were excluded. Participants who agreed to participate in the study were given oral explanations about the purpose of the study, and written informed consent was obtained during their hospital visits when coming for transfer to a treatment center. A convenience sampling technique was used to access patients on the day the researcher visited the hospital units before referral day. Data were collected by the project leader, a professional nurse with a master’s degree who is an established researcher. A face-to-face individual interview was conducted to determine the participants’ cancer-related experiences and quality of life. An in-depth interview allows individuals to express their experiences, perceptions, and attitudes about the topic under study without group influence [10]. The interviews were held in the unit consultation rooms because it provided a quiet environment. Open-ended and probing questions like “*How has undergoing cancer treatment been for you?*” and follow-up questions like “*How has it impacted your daily living?*” were asked. Interviews were audio-recorded and saved using an anonymized password. The interviews lasted between 30 min and 45 min. Data saturation was reached with the 9th participant from the sample size of eleven. However, the researcher continued with the interview even when no new information emerged.

### 2.2. Data Analysis

Data analysis is arranging, altering, and meaningfully summarizing the data [9]. Step 1: Transcription, Familiarization with the Data, and Selection of Quotations. The researcher listened to the audiotape several times and transcribed the data verbatim. Step 2: Selection of Keywords and labeled them as keywords. These keywords captured participants’ experiences directly derived from the data. Step 3: Coding. Short phrases or words, known as codes, were assigned to data segments that capture the data’s core message or theme. Step 4: Theme Development. The codes were organized into meaningful groups to identify patterns and relationships, thereby offering insights into the research question and presenting the core message of the theme. The themes developed were subjected to a triangulation process to ensure the researcher, supervisor, and co-coder’s validity and accuracy of the findings by cross-referencing with the literature review. Field notes were used to complement the coding process and all reached consensus of the developed themes.

### 2.3. Trustworthiness

Trustworthiness describes validating methods to analyze and represent qualitative have seedata [10]. A detailed audit trail documenting the research methods and processes was maintained to ensure trustworthiness [11]. Credibility was increased by rephrasing questions, repeating questions, or paraphrasing statements to ensure the credibility of information. Furthermore, follow-up clarity in participants where gaps were identified during listening to the tape and transcribing resulted in prolonged data engagement. Data transcripts were not shared with participants, but information was verified telephonically for clarity. Confirmability was ensured through co-coding; the supervisor and independent coder checked the transcripts, coded them, and reached a consensus with the project leader [11,12].

### 2.4. Ethical Consideration

Ethical approval was obtained from the Limpopo Provincial Department of Health, University Ethics Committee, protocol number SHS/15/PDC/02/1903, approved in January 2019. Institutional approval was obtained from the chief executive officers and managers of hospitals where the patients were admitted and receiving treatment. The hospital unit managers assisted the researcher with contacting potential participants, who were contacted and informed of the purpose of the study. Upon agreement, they were provided rom introduction section the flow andstudy information for informed consent. The information included their right to voluntary participation and a guarantee of anonymity and confidentiality during the study and reporting of findings. Participants were assured that if they felt overwhelmed during the study, they should report it and be referred for emotional support to alleviate discomfort. 

## 3. Findings

### 3.1. Participants’ Characteristics

The biographical information of the patients living with cancer (Table 1) showed that the mean age was 34 years, and their ages ranged from 33 years to 67 years. The majority were females, and the common cancers were mostly gynecological; one had head and neck cancer, while the others had pancreatic cancer and tIwo urogenital cancers. Among the patients, only two had comorbidities of hypertension and diabetes mellitus. Two males experienced urogenital cancers. Women were in the majority as they were more likely to seek medical help than men, especially among black communities, due to gender stereotypes and their perception of health services.

Below, Table 2 presents the findings from the in-depth face-to-face interviews with patients. The concepts and narratives are a presentation of inductive data acquired from participants. The findings were grouped into themes and sub-themes.

### 3.2. Theme 1: Experiences of the Burden of Bio-Physiologic Symptoms 

The signs and symptoms caused by cancer will vary depending on what part of the body is affected. Most patients undergoing cancer treatment will experience physical symptoms like pain, fatigue, skin changes, changes in bowel movement, difficulty in swallowing, unexplained muscle and joint pains, and other symptoms depending on the type of cancer treatment received, and some of these symptoms may continue even after their treatment is complete. 

#### Sub-Theme: Physical Distress

Pain and fatigue are common in all cancers and cancer treatments. Pain can drastically interfere with one’s quality of life by making it difficult or impossible to eat, sleep, and socialize.

Participants expressed their physical distress and reflections as follows:


*“I suffered pain, had pain medication, was not getting relieved, and I realized that the caregivers did not understand the intensity of the pain I suffered; I cried to myself until I cried loud to ask for more medication.”*
(#Uterine, age 52, F, stage 3)


*“In my journey as a cancer patient, during my chemotherapy period, I was so sick that I could not explain what my problem was, from pain, appetite, and weight loss. I wished someone could assess and identify the problem, which I could not explain.”*
(Cervix, age 48, Stage 2)

A participant said:


*“My whole body was so sore……. dry and itchy, and my skin looked like I was struck by lightning; my completion was gone.”*
(# Breast, Age 45, F stage 2)

She further said the following: 


*“I do not know what to eat……, nothing stays in my stomach, sometimes I have diarrhea, and at times I feel constipated, and this makes me reluctant to eat.”*
(Pancreas, Age 33, F, Stage 2)

Another participant said the following: 


*“My mouth is always dry, nothing is palatable, my gums were sore, and I lost some of my teeth.”*
(Esophagus, Age 68, M, Stage 4)

A different participant said the following:


*“I know that there was nothing that the doctors and nurses could have done; it is only that I needed a lot of support on this issue because it was not only my hair; even my nails got very dark, and I was shy to show my hands.”*
(# Breast, age 45, F, Stage 2)

Cancer cells are destroyed during chemotherapy and radiation treatment, and normal cells are also affected. The side effects of cancer treatment cause patients to experience feelings of loss of control, having a direct and negative impact on how they cope with cancer and their psychological well-being.

### 3.3. Theme 2: Experiences of Feelings of Uncertainty, Anxiety, and Emotional Pain

Uncertainty is common in cancer patients’ journeys. It can cause them to experience prolonged feelings of a loss of control and anxiety, which negatively impact how they cope with cancer and QoL.

#### Sub-Theme: Emotional Distress

Cancer diagnosis brings along uncertainty, and participants feel anxious or frightened about whether treatments will work and what will happen in the future.

A participant in the interview indicated how she has lived with uncertainty in the following statement:


*“I did not know if I would get healed or would die from the cancer; most of the people I know with cancer have died. I received counseling from the nurses, but in real fact, I was not satisfied; I needed to hear it from other people who have had the same condition, had undergone the same treatment.”*
(# Prostate, Age 58, M, stage 3)

A participant said the following: 


*“I feel so alone even if there are people with me. I wish I could share my fears and insecurities with my family.”*
(Breast age 45, F, stage 2)

Emotional stress is common for people with cancer, and they worry about the future, whether treatments will work, and what will happen if the cancer returns, resulting in decreased quality of life.

One participant said the following:


*“I know that there was nothing that the doctors and nurses could have done; I only needed a lot of support on this issue because it was not the hair only; even my nails got very dark, and I was shy to show my hands.”*
(# Cervix, Age 48, F, stage 3)

A participant said: 


*“I am not well because I am always wet, draining some fluid, so I need to have a towel, not stain the clothes, and I feel smelly myself, so I am not free to go out.”*
(Uterine, age 52, F, stage 3)

Another participant supported this statement and said the following: 


*“My skin colour changed, I was yellow, my eyes…… eeh… when looking at the mirror, your heart just broke and felt empty.”*
(Pancreas, Age 33, F, Stage2)

Emotional distress is common in patients throughout the cancer journey and causes unpleasant experiences that are emotional, social, and spiritual. Participants in the study experience prolonged feelings of loss of control, feelings of anxiety, and uncertainty, which negatively impact how they cope with cancer and QoL.

### 3.4. Theme 3: Experiences of Lack of Social Support

The cancer treatment journey takes time and affects social and economic conditions. It can lead to poorer subjective well-being of patients and affect the quality of life regarding regular treatment follow-up, additional therapeutic needs, and financial obligations. 

#### Sub-Theme: Social Distress

Cancer diagnosis and treatment present challenges in numerous areas of social life, including family life, relationships at the workplace and fear of income, which causes social distress and impacts on the quality of life.

One participant explained the following: 


*“Even if I were to tell, they would not understand as they have not had the experience. I believe having more support from those who share the experience and fears would be better.”*
(# Breast age 44, F Stage 2)

Another participant said the following:


*“You feel family and friends would not understand and would be unable to answer my questions or allay my anxieties.”*
(# Breast age 54, stage F, stage 3)

A different participant said the following:


*“I know that there was nothing that the doctors and nurses could have done; I only needed a lot of support on this issue because it was not the hair only; even my nails got very dark, and I was shy to show my hands.”*
(# Cervix, Age 48, F, stage 3)

One patient expressed concerns in the following manner: 


*“Now that I have cancer, as a single parent, will I be able to continue with my work? How will I explain my absence from work when I have to undergo treatment, and sometimes, I feel weak and unproductive.”*
(# Cervix, Age 49, F, stage 3)

She further added the following: 


*“I have a problem with my family that if I die from this diagnosis, who is going to take care of my children as they were in primary school.”*
(# Cervix, Age 49, F, stage 3)

Another participant explained the following: 


*“I felt that cancer was heavy to absorb, could not talk about it even to my children, needed to keep it close till I was ready. Nurses spoke about this openly about my diagnosis of my cancer amongst themselves even in front of non-concerned; I felt worthless.”*
(Uterine, Age 52, stage 3)

Another participant explained the following:


*“I needed to be with my older relative or my wife when I received the news. It took me time to inform them, and that delayed my time to agree to start treatment.”*
(# Prostate, Age 67, M, stage 3)

Social distress affects the social relationships, social activities, and financial problems of cancer patients. It may positively or negatively contribute to how they cope and respond to the cancer diagnosis, prognosis, and treatment. However, most rural patients suffer due to a lack of social and financial support. 

### 3.5. Theme 4: Experiences of Maladaptive Behaviors

Psychological distress is a feature of impaired mental health or common mental disorders, including depression or anxiety, influenced mainly by the subjective experience of cancer burden. 

#### Sub-Theme 1.4. Psychological Distress

Psychological distress refers to non-specific symptoms of stress, anxiety, and depression associated with normal fluctuations of mood in most people. The diagnosis of cancer and its treatments result in affective disorders and changes in self-esteem and self-worth, which may result in depression.

One participant indicated the following: 


*“With the stress of having cancer, I resorted to alcohol, not eating; I did not care what happened next; I thought it was the end of me.”*
(# Prostate, Age 58, M, stage 3)

This participant further indicated: 


*“I thought that it is best to commit suicide instead of dying like my friend who suffered a lot of undergoing treatment at the provincial hospital.”*
(# Prostate, Age 58, M, stage 3)

Another indicated the following: 


*“With the stress of having cancer, I thought it was the end of me; I would rather die than experience the feeling of discomfort and resorted to self-medications, using over-the-counter medicine and mixtures of herbs from the herbalist.”*
(# Cervix, Age 47, F, stage 2)

Another participant reiterated the following: 


*“It was no use to keep my disease a secret because my sexual libido has decreased. My manhood has been removed, so sharing my frustration might help; maybe my wife may understand my present state.”*
(# Prostate, Age 58, stage 3)

Self-esteem is a fundamental psychological resource that is highly associated with health-related behaviors, psychological well-being, and mental health, which can cause cancer patients to take time to disclose their diagnosis and, due to certain fears, use avoidant coping mechanisms to conceal their feelings and safeguard their well-being, resulting in the sense of exclusion and reduced self-expression.

### 3.6. Theme 1.5: Experience of Beliefs in Cultural Care 

Religion is a cultural system of beliefs, practices, rituals, and symbols designed to help individuals with sacred aspects of one’s life. Spirituality is a personal search for answers concerning the meaning of life.

#### Sub-Theme: Cultural Beliefs, Norms, Values, and Spiritual Care 

The modern healthcare system is built on Western values and ideologies and does not necessarily ensure cultural competence in healthcare professionals, which may impact patient care. Cultural competence is one of the greatest challenges healthcare faces today.

A participant reflected on her experience in the following:


*“I am a person that I would normally like my stories not to be told casually to people. I don’t know; my vaginal bleeds are private. Examination by a male doctor with a male nurse in the room was something that made me feel like I was walking naked in public. Those two are very young, maybe younger than my first grandchild.”*
(# Uterine, Age, 52, F. Stage3)

Spiritual care for cancer patients is an important part of person-centered care, and it can help patients cope with their illness. 

One participant expressed the following:


*“I forcefully requested for discharge from the hospitals to go for the spiritual and traditional healers and ended up signing a refusal of hospital treatment before I could agree to treatment. I was angry and could not believe and accept the cancer news.”*
(Esophagus, Age 68, M, stage 3)

Recognizing and validating an individual’s spiritual needs can allow the medical team to connect more deeply with a patient and help maximize support from within an individual, their close circle, and the greater community.

## 4. Discussion

The findings from the in-depth interviews highlighted that, indeed, participants experienced symptomatic distress of pain, fatigue, loss of appetite, skin irritation, and mood swings. These symptoms are prevalent in patients undergoing cancer treatment as cancer treatment also affects normal cells and tissues, disrupting the normal functioning of the various cells and organs. Inadequate management of treatment toxicities can severely affect a person’s quality of life and lead to increased morbidity and mortality [13]. A comprehensive assessment of the side effects is essential to identify risks and manage toxicities early; however, in rural Limpopo, a comprehensive assessment is lacking as the district and regional hospitals lack oncology health professionals, such as oncology nurses and patient navigators. For example, the effects of treatment on the skin are unbearable to manage as patients experience scorching temperatures of above 40 degrees and a lack of water and sanitation. Poor access to clean water and proper sanitation can be distressing and affect the person’s quality of life due to related waterborne diseases and the effects of cancer treatment. Another participant in an advanced stage of uterine cancer endured pain that interfered with their quality of life by making it difficult or impossible to eat, sleep, and socialize; the inability to maintain adequate perineal hygiene and toiletry resulted in physical distress. Inadequate management of treatment toxicities can severely affect a person’s quality of life and lead to increased morbidity and mortality.

Cancer significantly increases the risk of developing depression and low self-esteem related to changes in body image. Low self-esteem is associated with a high frequency and intensity of symptoms associated with many mental health disorders, such as denial or avoidance of the situation. Cancer patients require adjustments and coping techniques to be provided to them at an early stage of the disease, which is impossible in rural settings where health-seeking behavior is often delayed due to a lack of knowledge and poor health education and health promotion interventions for chronic conditions. Niveau et al. [14] suggest that special attention should be paid to specific groups of patients, such as young adult patients or those who experience significant physical damage, in whom cancer may cause a decrease in self-esteem and should be identified early and provided with intervention therapies targeting the self-perceptions to promote psychological adjustment to cancer.

Participants in the study expressed emotional distress related to emotional pain, anxiety, and uncertainty about treatment and the prognosis of the disease; they alienated themselves out of fear of stigma and discrimination and people feeling sorry for them, and they drowned in their pain and sorrow, as participants indicated skin color changes. Changes in body image can bring about emotional distress in patients since people are concerned about how they look, and these can lead to social isolation, feelings of shame, and fear of stigmatization. In rural communities, very dark-pigmented people are regarded as witches. Therefore, body image can be quite a distressing factor. Brederecke et al. [15] reported that women with breast and gynecological cancers reported diminished levels of self-acceptance scores compared to women with visceral cancers. This may be related to femininity (personal appearance) for other races, but for rural black African women, being feminine is associated with childbearing capabilities above anything else, and women may feel incomplete and may not even be confident in themselves and fear stigmatization. Furthermore, Brederecke et al. [15] contend that diminished body image satisfaction can have negative consequences on physical and psychological health and interpersonal relationships, leading to psychological distress, and thus impair quality of life.

Participants expressed uncertainty, a loss of hope for the future, and a fear of death as they witnessed some cancer patients dying during the cancer treatment. Participants felt they needed support from other cancer patients who have undergone the same treatment as them to be hopeful. The loss of hope has detrimental effects on how an individual perceives and values life, and some may respond negatively, as one participant indicated, by resorting to alcohol. Dekker et al. [16] indicate that emotions are maladaptive. When they are severe or persistent, they interfere with functioning, leading to the avoidance of cancer treatment, depression leading to unrealistic negative expectations and a lack of motivation to continue treatment, or fear of death leading to unrealistic optimism regarding the outcome of further treatment. Although some patients can cope with the emotional burden of cancer, emotional support is required as some participants yearn to be motivated by those who rose above the cancer diagnosis.

Numerous studies have linked clinical uncertainty with diminished QoL and resilience, anxiety, depression, and other adverse effects [15,16,17]. Participants indicated that health professionals should provide them with more information, while others felt that those who have beaten the cancer could provide hope for them. Using other patients as a source of emotional support relies on personal and individual experiences and resilience, which can sometimes be emotionally draining for the survivor. Slevin et al. [18] indicate that it is helpful for a newly diagnosed patient to talk to someone who has been through similar trauma. However, it can also be stressful; patients who measure their progress using other patients doing well may become distressed if they do not adjust favorably. Other means of providing emotional support include online self-management support, patient discussion, support groups, and expert volunteers [16,19].

Cancer diagnosis and treatment bring about changes in patients’ paths of life, in their daily activities, work, relationships, and family roles, and it is associated with a high level of patient psychosocial stress. Usta [20] indicates that social support consists of people we can count on to provide ongoing emotional support, affirmation, information, and assistance, especially in times of crisis. Kyriazidou et al. [21] indicate that patients with strong social support develop more optimistic feelings, which enable them to increase their confidence and hope to fight and cure cancer successfully.

Participants expressed feelings of loneliness and longing to share their feelings with loved ones, but they were reluctant because they would not understand the social distress they experienced. Participants were also anxious about the financial security of their children in case they pass on, the workplace relationships when they could perform adequately, and their difficulties in communicating their diagnosis to managers. Disclosing an illness and frequent sick leaves can be socially distressing for participants, especially with the high rate of unemployment in the country. de Rijk [22] reported in their study that cancer survivors reported that employees were not supportive and were not always optimistic about the role of being supportive as employers. They perceived barriers related to support, communication, work environment, discrimination, and perception of ability to work. Cancer treatment causes fatigue and exhaustion, resulting in one being unproductive and impacting on the quality of life in performing activities of daily living (Rodriguez-Gonzalez et al. [23], and it also restricts the person’s physical and social activities and their ability to return to work and be economically productive [24]. Moreover, rural patients may not be professional and highly skilled, which places them at risk of losing their employment or having medical insurance as unskilled personnel.

Social relationships can provide emotional connections, security, reassurance, and guidance. A male participant only disclosed the cancer diagnosis to his next of kin when there was evidence of symptoms that he could not conceal. Disclosure is not a one-time event but rather an ongoing process, with choice and privacy being fundamental. Woodgate et al. [24] indicate that there is partial and active open disclosure that may be triggered by changes in physical appearance, such as hair loss or scars, changes in mood or speech, or physical limitations, and active open disclosure reveals their true selves, particularly to those they trust. Corovic et al. [25], Rodriguez-Gonzalez et al. [23], and Katsaros et al. [26] indicate that family members offer emotional support like esteem, trust, concern, and listening. In contrast, instrumental support consists of money, and cancer patients do not receive social grants, which is a burden for those lacking financial security as they may lack transportation to return visits for consultation, treatment, and follow-up.

Cancer patients experience mental distress, which may lead to maladaptive behaviors such as anger, aggression, suicidal tendencies, and even self-medication and discontinuation of treatment or refusal of treatment. Psychological distress can range from mild emotional stress to serious psychological distress that may be overwhelming. A participant in the study indicated that she felt worthless when healthcare professionals were discussing her condition, discussing her as if she were not around, which is psychologically distressing; communication should be culture-sensitive, and health professionals should understand diversity in communication. The finding further revealed that the participant resorted to signing a refusal of treatment form to consult the spiritual traditional assistance since she had lost hope with the current treatment. Denial results from psychological stress, and at the most, men might be more likely to deny during the terminal phase [27], like in the case of the participant who was already in the advanced stage (stage 3) and needed to find the natural cause of the disease due to the spiritual belief. Furthermore, the authors indicate that black patients, however, are more likely to rely on denial of affect, whereas whites display a more behavioral escape [28]. The findings further indicate the participants’ indulgence in alcohol, the use of over-the-counter medication, and the thought of suicide. Kreitler [29] indicates that denial may interfere with treatment, disrupt the process of assimilating the stressful event, adversely affect interpersonal relations, and constitute cumulative stress or depression, even immunocompetence.

Cultural care involves understanding and respecting patients’ and their families’ cultural beliefs, practices, and values. Rural communities rely on the Indigenous knowledge system, and a participant reported how embarrassed a male practitioner was examining her; she felt that her dignity was taken away since it is culturally unacceptable and morally wrong. Niveau et al. [14] indicate that self-esteem is a fundamental psychological resource highly associated with health behaviors, psychological well-being, and mental health. Sexual and reproductive cancers cause severe mental distress because, culturally, older women cannot undress in front of young male practitioners; it is disrespectful and frowned upon. The findings of this study are supported by those of Indongo and Robert [30], that the healthcare system’s work culture causes ethical uncertainty or moral burden for patients and families. Participants have their own belief systems, and they are different. Understanding their belief systems and acknowledging that they develop over time, healthcare professionals must support cancer patients throughout the cancer trajectory. Unfortunately, cancer services are ineffective and uncoordinated in rural settings as most patients fall through the cracks or die at home.

Healthcare professionals should also recognize such behavior and respect individual cultural beliefs as these patients are experiencing a critical period and are nearing the end of life. Fernandez-Feito et al. [31] also highlight the importance of recognizing the spiritual needs of participants through active listening, normalizing conversations about their beliefs, and respecting silences, which are successful communication strategies for providing spiritual care. Furthermore, Derya et al. [27] indicate that nurses providing patient care should be sensitive to cancer patients’ perceptions of the disease and coping methods in the cancer patient journey.

## 5. Implication for Practice

The findings provide valuable insights into the biopsychosocial concerns of patients living with cancer. The health service system must appreciate the problems and needs of this population and provide holistic and comprehensive health education regarding the side effects of the treatment and when to seek medical help.

Healthcare professionals should be able to provide social support by building a therapeutic relationship with patients for continuity of care by improving the levels of care from Primary Health Care (PHC) to tertiary care at the provincial level.

Provision of supportive resources beyond social support to improve health behaviors and cancer survivorship outcomes.

The healthcare professionals’ curriculum should include in-depth transcultural training on the influences of culture on patient care.

The government should improve cancer survivorship, reduce health disparities, and promote health equity.

### Limitations

The limitation was exploring distress among patients with different cancers, including all stages of the disease and all treatment approaches, ensuring its representativeness. The study had a small sample, and the findings cannot be generalized but highlight the plight of rural communities lacking access to quality care equity.

## 6. Conclusions

The findings from this qualitative study provide additional evidence of the lived experiences regarding the bio physiologic symptomatic distress, emotional distress, psychological distress, and the use of indigenous health beliefs. Patients are unique and experience distress differently, especially among rural communities where cancer is still labeled as a curse or stigmatized. The healthcare system should acknowledge traditional and spiritual interventions and be culture sensitive. A multisectoral approach is required from healthcare professionals, allied health professionals, support structures of social workers and clinical psychologists, and family members to understand and fulfill these needs. Comprehensive family-centered care is crucial to improving the quality of life.

## Figures and Tables

**Table 1 curroncol-32-00043-t001:** Demographic profile of the patient participants.

Hospital	Age	Gender	Site of Cancer	Stage of Cancer	Treatment Modality
A	54	F	Breast	Stage 3	Chemo/Radiotherapy
	47	F	Cervix	Stage 3	Chemo/Radiotherapy
B	33	F	Pancreas	Stage 2	Chemo/surgery
	58	M	Urogenital	Stage 3	Chemo/Radiotherapy
C	48	F	Cervix	Stage 2	Chemo/Radiotherapy
	68	M	Esophageal	Stage 4	Chemo/Radiotherapy
	45	F	Breast	Stage 2	Chemo/Radiotherapy
D	54	F	Breast	Stage 2	Chemo/Radiotherapy
	49	F	Cervix	Stage 3	Chemo/Radiotherapy
E	67	M	Urogenital	Stage 3	Chemo/Radiotherapy
	52	F	Uterine	Stage 3	Chemo/Radiotherapy

**Table 2 curroncol-32-00043-t002:** Findings from in-depth individual interviews.

Theme	Sub-Theme
Experiences of the burden of bio-physiologic symptoms	Physical distress
Feelings of uncertainty, anxiety, and emotional pain	Emotional distress
Experiences of lack of social support	Social distress
Experiences of maladaptive behaviors	Psychological distress
Experience of beliefs in cultural care	Cultural beliefs, norms, values, and spiritual care

## Data Availability

All data are presented in this manuscript.

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
