# Peer review of "Psychosocial Distress and the Quality of Life of Cancer Patients in Rural Hospitals in Limpopo Province: A Qualitative Study"

_curroncol, 2025, doi:10.3390/curroncol32010043_

Round 1

Reviewer 1 Report

Comments and Suggestions for Authors

To authors,

I have some suggestions to improve the quality of qualitative research reporting. Let me share some thoughts on the methodology, results and discussion section.

Methodology

1) Describe in detail how participants will be sampled

2) Present the data collection period

3) Did you conduct any repeat interviews?

4) Was there a process for returning transcripts?

5) Describe interviewer qualifications, experience and training of researchers

6) Number of data coders?

7) A step-by-step description of the data analysis process would be nice (e.g., coding tree or tables and figures).

8) Was there a process of participant checking of the finalized findings?

Results and discussion

1) Some sub-themes are redundant, and the naming of the Theme is not sufficiently representative of the sub-themes.

2) Themes are too broad to be considered specific descriptions of phenomenological characteristics.

3) The themes that emerged are not specific to the rural population and it is difficult to find new meanings that go beyond the known categories of cancer patients' problems.

4) Presenting limitations in terms of the rigor of qualitative research 

That's all for now.

Yours sincerely, Reviewer.

Author Response

Please find the responses in the attachment.

Reviewer 2 Report

Comments and Suggestions for Authors

The article presents a relevant and highly scientifically interesting topic, particularly in the area of cancer care in rural contexts, which is an underexplored area in many studies. It comprehensively addresses the biopsychosocial, emotional, and cultural aspects affecting cancer patients, giving it significant value. However, the manuscript’s quality could be improved in terms of structure, clarity, and depth in integrating the results with existing literature. Below, some aspects are specified that should be considered and modified for its publication in the journal.

The INTRODUCTION presented is well-structured and addresses a relevant topic within the field of oncology, particularly in the context of low- and middle-income countries. However, the following aspects should be considered:

-        Dispersed focus: While the introduction covers various aspects of the impact of cancer, it could benefit from a more precise focus. For instance, grouping the physical, psychological, social, and spiritual effects under a clear framework would help avoid redundancies.

-        Greater local contextualization: Although the article focuses on rural hospitals in Limpopo province, the introduction does not provide specific information about this context. What particular characteristics of this region exacerbate the burden of cancer? What types of oncology services are available or lacking? Including this information would better connect the global problem to the local setting.

-        More clearly highlight the gap in the literature that the study aims to address.

-        Reformulate the objectives to make them more specific and measurable, avoiding general statements.

-        Repetitive writing: Certain assertions are reiterated, such as the vulnerability of patients to emotional and social changes or the interference of cancer with quality of life. A more concise approach could improve the text's flow.

The design and METHODS are well-defined and appropriate for the study's qualitative and exploratory approach. However, several areas for improvement have been identified:

-        While it is mentioned that 11 participants were interviewed and saturation was reached with the ninth, there is no detail on how patients were initially selected (e.g., was purposive sampling, convenience sampling, or selection based on specific criteria used?). Providing a description of the inclusion and exclusion criteria would strengthen this section.

-        Although it is stated that themes were triangulated, it is not explained who participated in this process (e.g., researchers, external experts) or how it was conducted (e.g., cross-referencing with previous literature or validation by participants).

-        While the qualitative analysis process is described in general terms, the specific analytical approach used is not specified. Was thematic analysis employed?

-        Although saturation is said to have been reached with the ninth participant, there is no explanation of why an initial sample size of 11 participants was deemed appropriate or how saturation was assessed. A more robust discussion of this topic would enhance the methodology.

The RESULTS section presents relevant findings that are coherently organized, grouped into main themes reflecting the experiences of cancer patients in rural hospitals. However, while the presentation of data is solid and structured, there are areas that could be improved to enhance clarity, analytical depth, and the academic utility of the work. The following suggestions are outlined:

-        Organization: Structure the results around clearly defined thematic categories (e.g., physical, emotional, social, and cultural aspects). Use subheadings to divide the main topics and facilitate readability.

-        Depth: Provide more detail on how the main themes were identified. Include additional information on differences among participants (e.g., based on gender, type of cancer, or stage of treatment).

The DISCUSSION AND CONCLUSIONS section comprehensively addresses the biopsychosocial, emotional, and cultural challenges faced by cancer patients, particularly in rural settings. The analysis is well-supported with relevant references, but there are areas requiring greater clarity, structure, and connection between the findings and practical implications. To improve this section:

-        Connection to results: Link the findings more directly to the thematic categories presented in the results. Additionally, explain how each finding addresses the study objectives.

-        Comparison with previous studies: Systematically compare the findings with prior research, highlighting similarities and differences. Discuss how the study contributes to existing knowledge and what aspects are novel.

-        Practical implications: Provide specific recommendations for healthcare systems, professionals, and public policy.

-        Study limitations: Explicitly acknowledge the study's limitations (e.g., sample size, focus on a specific region). Suggest how future studies could address these limitations.

-        Conclusion: Reformulate the conclusion to synthesize the main findings and their relevance. Avoid generalizations and limit conclusions to what the data can support.

With these revisions, the manuscript will not only be scientifically stronger but also more appealing to readers and researchers in the fields of oncology and public health.

Author Response

(The authors gave the same response as above.)

Reviewer 3 Report

Comments and Suggestions for Authors

1.What kind of qualitative study does this research use? Please present a more complete title, such as Ethnographic Research Method, Narrative Method, Phenomenological Method, Case Study Research, Grounded Theory Method, Focus Groups….

2. I'm curious about the authors' decision to use the qualitative study method instead of the quantitative one. Could you provide some insights into this choice?

3.The authors listed several themes, but how did the relative importance of these themes be evaluated?

4. For the discussion section, did any results differ from the past literature? Or could special results in the study that are different from the past literature be discussed?

5. The conclusion section seemed not to have clear or concrete results descripted.

Author Response

(The authors gave the same response as above.)

Round 2

Reviewer 1 Report

Comments and Suggestions for Authors

Thank you for your hard work in revising your paper. 

There are no further comments.

Reviewer 2 Report

Comments and Suggestions for Authors

The authors have taken the suggestions into account. They have done a good job. I think the manuscript is ready for publication.

Reviewer 3 Report

Comments and Suggestions for Authors

OK